# Optical Microfiber All-Optical Phase Modulator for Fiber Optic Hydrophone

**DOI:** 10.3390/nano11092215

**Published:** 2021-08-28

**Authors:** Minwei Li, Yang Yu, Yang Lu, Xiaoyang Hu, Yaorong Wang, Shangpeng Qin, Junyang Lu, Junbo Yang, Zhenrong Zhang

**Affiliations:** 1Guangxi Key Laboratory of Multimedia Communications and Network Technology, School of Computer, Electronics and Information, Guangxi University, Nanning 530004, China; 1913301020@st.gxu.edu.cn (M.L.); 1913391019@st.gxu.edu.cn (S.Q.); 1913392039@st.gxu.edu.cn (J.L.); 2College of Liberal Arts and Sciences, National University of Defense Technology, Changsha 410073, China; 2019111068@bupt.edu.cn (Y.W.); yangjunbo@nudt.edu.cn (J.Y.); 3State Key Laboratory of Transducer Technology, Shanghai Institute of Microsystem and Information Technology, Chinese Academy of Sciences, Shanghai 200050, China; 4College of Meteorology and Oceanography, National University of Defense Technology, Changsha 410073, China; shangshan_101@126.com (Y.L.); huxiaoyang08@sina.cn (X.H.)

**Keywords:** optical microfiber, PGC demodulation, fiber optical hydrophone, all-optical

## Abstract

In order to meet the needs of phase generated carrier (PGC) demodulation technology for interferometric fiber optic hydrophones, we proposed an optical microfiber all-optical phase modulator (OMAOPM) based on the photo-induced thermal phase shift effect, which can be used as a phase carrier generation component, so as to make the modulation efficiency and working bandwidth of this type of modulator satisfy the requirements of underwater acoustic signal demodulation applications. We analyzed the modulation principle of this modulator and optimized the structural parameters of the optical microfiber (OM) when the waist length and waist diameter of OM are 15 mm and 1.4 μm, respectively. The modulation amplitude of the modulator can reach 1 rad, which can meet the requirements of sensing applications. On this basis, the fiber optical hydrophone PGC-Atan demodulation system was constructed, and the simulated underwater acoustic signal test demodulation research was carried out. Experimental results showed that the system can demodulate underwater acoustic signals below 1 kHz.

## 1. Introduction

With the development of optical fiber sensing technology, various types of optical fiber sensors have the advantages of simple manufacturing, low cost, and small size. Therefore, they show good application potential in the field of ocean exploration [1,2]. Additionally, the hydrophone system is an important instrument of the marine monitoring system and the monitoring network [3]. Compared with traditional electric hydrophones, fiber optic hydrophones have the advantage of resistance to electromagnetic signal interference, higher sensitivity, simple construction, and convenience for large-scale multiplexing into arrays. Additionally, the underwater system can be all-optical without electricity. The hydrophone system is in a period of comprehensive transition from electrical equipment to all-optical equipment [4,5]. According to different sensing principles, fiber optic hydrophones can be divided into interference type and fiber-grating type [6]. Interferometric hydrophones can be divided into different structures of interferometers: the Mach-Zehnder interferometer, Michelson interferometer, Fabry–Perot interferometer, and Sagnac interferometer. Among them, the Michelson interferometer hydrophone is widely used in various underwater acoustic environment monitoring systems because of its simple structure, high sensitivity, and easy array formation.

The fundamental working principle of the Michelson interferometric hydrophone is that the external vibration signal causes the phase change of the arm of the interferometer. At present, the demodulation methods of interferometric fiber optic hydrophone mainly include: the homodyne demodulation method [7], heterodyne demodulation method [8] and 3 × 3 diversity detection method [9,10]. PGC demodulation technology is a kind of passive homodyne demodulation method, which introduces a certain high frequency and large amplitude phase carrier modulation signal with wider bandwidth than the detection signal into the optical fiber hydrophone [7].

According to the different loading methods of the carrier modulation signal, the demodulation system can be divided into two methods to implement modulation: external modulation method and light source internal modulation method. The external modulation method cannot ensure that the underwater part of the hydrophone system is all-optical. Therefore, it is necessary to develop an all-optical phase modulator with carrier generation capability. As one of the core components of the optic fiber hydrophone system, the modulator directly determines the key technical indicators of the hydrophone system such as signal-to-noise ratio (SNR), low-frequency detectability, and signal demodulation processing methods. At present, most optical phase modulators are based on electro-optical effects [11], thermo-optical effects [12], and elastic-optic effects [13]. These several modulators require electrical signals and have an enormous size, which cannot be applied to the hydrophone unit. To meet the application requirements of optical systems such as a new generation of large-scale optical communication networks, optical sensing, and optical information processing, many all-optical modulators and optical switches have already been developed [14,15,16,17,18,19].

The all-optical phase modulator using silicon-based materials has a large modulation bandwidth and is expected to be applied to a new generation of integrated communication networks [20,21]. However, it has problems such as a complex structure, high power consumption, high cost, and poor interconnection compatibility with the optical fiber system. In particular, its phase modulation depth cannot meet the need of PGC demodulation systems. The fiber optic hydrophone system is mainly used to detect low-frequency underwater acoustic signals below 1 kHz. According to the principle of the PGC demodulation system, the modulator needs to be able to generate a carrier signal with a bandwidth of 10 kHz, and the amplitude of the modulated signal within the bandwidth needs more than 1 rad. In addition, although the all-optical phase modulation technology based on the integration of graphene and optical fiber is easy to integrate with the optical fiber sensing system and has a large modulation bandwidth, its modulation depth has difficulty meeting the requirements of hydrophone applications [22,23].

OM is a typical sub-wavelength diameter waveguide, which can confine the optical field to the wavelength scale or sub-wavelength scale, realize micro-scale and nano-scale photon transmission and manipulation, and exhibit good photon control and integration characteristics [24,25,26]. In addition, it is easy to integrate well with existing optical systems (due to the residual fiber tail) and has become a great carrier for the development of new functional devices. For example, Zhang et al. wound OM on an elastic rod and used the contraction of the rod to change the refractive index of the optical fiber to form an OM phase modulator based on the elastic-optic effect. However, it needs a higher voltage to drive, and the size is too large for practical application.

What has caught our attention is that the light absorption heating effect is common in all kinds of optical fibers, and OM is no exception. This will cause the thermal expansion of the optical fiber material and the change of the refractive index [27], leading to the phase change of the OM, the photo-induced thermal phase shift effect. Leading this kind of phase shift into the interferometer will cause a change in the intensity of the interference light. OM has greater transmission loss than ordinary communication fibers, and OM has strong light confinement and great light absorption capabilities. Therefore, an all-optical phase modulator based on the photo-induced thermal phase shift effect can be developed [28]. To further optimize the performance of the modulator, we studied the influence of the waist diameter and waist length of OM on the phase response amplitude. An all-optical phase modulator that can generate carrier waves for PGC demodulation systems is proposed. Additionally, the demodulation test research on the simulated underwater acoustic signal was carried out.

## 2. Simulation Results

The OM used in this paper is made by a single-mode fiber through the flame scanning method [29,30]. Its structure is mainly divided into three parts: conventional fiber tail, tapered transition area, and uniform waist area, as shown in Figure 1a.
$r_{w}$ is the radius of the uniform waist area.

### 2.1. Affects of OM’s Uniform Waist Area Diameter

The photo-induced thermal phase shift of OM can refer to the phase change Δφ of the single-mode fiber [27].
(1)$$\Delta \varphi =\frac{2\pi }{\lambda }\frac{\partial n}{\partial T}\frac{\eta }{\rho C_{v}}\int_{0}^{L}\int_{0}^{\infty }\alpha_{p}\left(r,z\right)w_{p}\left(r,z\right)f_{s}\left(r\right)2\pi rdrdz$$
where λ is the wavelength. ρ is the density of the fiber, and $C_{v}$ its specific heat. η is the fraction of pump energy absorption turned into heat. ∂n/∂T is the index of refraction, which can be affected by heat. Some parameters are related to the azimuthal position, where r is the radial position, and z is the position along the fiber. $\alpha_{p}\left(r,z\right)$ is the absorption coefficient of the fiber. $w_{p}\left(r,z\right)$ is the pulse energy density of pump. $f_{s}\left(r\right)$ is the signal mode intensity of the fiber section.

As $\lambda_{s}$ = 1550 nm, relevant physical parameters of fiber (∂n/∂T = 1.1 × 10^−5^, ρ = 2.2 × 10^3^ Kg/m^3^, $C_{v}$ = 741 J/Kg/K) can be substituted into Equation (1) to simplify and calculate; Δφ can be expressed as [27]
(2)$$\Delta \varphi_{i}\left(rd\right)=4.28\times 10^{-11}\frac{\eta }{\lambda_{s}}\frac{E_{abs}}{A_{eff}}$$
where $E_{abs}$ is the energy absorbed by the fiber. $A_{eff}$ is the effective interaction area, which involves spatial overlap integrals between the pump spot, signal spot, and optical fiber absorption section,$A_{eff}$, and is defined as [27]
(3)$$A_{eff}=\pi \left(r_{s}^{2}+r_{p}^{2}\right)\frac{1-\exp\left(-\frac{s^{2}}{r_{p}^{2}}\right)}{1-\exp\left(-\frac{s^{2}}{r_{s}^{2}}-\frac{s^{2}}{r_{p}^{2}}\right)}$$
where $r_{s}$ is the radius of the signal light spot, $r_{p}$ is the radius of the pump light spot, and s is the fiber cross-sectional radius. According to Equation (2), the phase response amplitude can be improved by increasing $E_{abs}$ or decreasing $A_{eff}$. The increase of $E_{abs}$ is related to the absorption coefficient of the optical fiber itself. The optical absorption coefficient of doped fiber is larger but under the modulation of larger modulated light. It is difficult to use the photo-induced thermal phase shift effect of ordinary doped fiber for phase modulation, and the modulation efficiency is low [27]. In order to achieve low power consumption in a practical application, the output power of the pump light should not be increased excessively. Thus, the phase response amplitude is increased by decreasing $A_{eff}$.

OM has strong light confinement capability [31,32], which can constrain light in the uniform waist area of a micron scale. The cross section of the uniform waist area is filled with light, so it is generally believed that the photo-induced thermal phase shift effect of OM mainly occurs in the uniform waist region. We propose that the optical fiber absorption area section can be equivalent to the uniform waist area section of OM. In Equation (3), we set $r_{s}$, $r_{p}$, and s to the same value as $r_{w}$ in the calculation. According to Equations (2) and (3), the relationship between the phase response value of OM and the waist radius is shown in Figure 1b. It can be seen that the phase response of the photo-induced thermal phase shift effect increases with the decrease of the uniform waist area radius of OM. When the radius of OM’s uniform waist area decreases to about 0.48 μm, the phase response decreases rapidly. Therefore, the part ($r_{w}$ ≥ 0.48 μm) before the rapid decrease of the OM phase response of the photo-induced thermal phase shift effect was primarily analyzed.

The finite element method (FEM) was used to simulate the mode field energy distribution of the OM section with different cross-sectional areas. The section diameters $d_{w}$ were set as 1.8 μm, 1.6 μm, and 1.4 μm, respectively. The mode field distribution of the OM section was shown in Figure 1c–e. It can be seen that about 80% of the energy is confined to the section of the uniform waist area. Moreover, the strong light confinement capability of OM makes it possible to synchronously reduce the spot area when reducing the waist region diameter, thus realizing the compression of the energy density of the light field. Relevant simulations are also carried out. When the diameter of the uniform waist area of OM is close to the extreme point of $r_{w}$ ($r_{w}$ = 0.48 μm), taking the diameter of 1.0 μm as an example, more energy has been propagated outside the fiber, as shown in Figure 1f. It is concluded that the response amplitude of the OM phase modulator can be improved by decreasing the diameter of the OM uniform waist area within a range.

### 2.2. Affects of OM’s Uniform Waist Area Length

In the OM drawing process, the volume conservation principle is conserved, and the relationship between the radius of uniform waist area $r_{w}$ and the initial radius $r_{0}$ satisfies the following equation [33]:(4)$$r_{w}\left(x\right)=r_{0}\exp\left[-\frac{1}{2}\int_{0}^{x}\frac{dx^{\prime }}{L\left(x^{\prime }\right)}\right]=r_{0}{\left[1+\frac{\alpha x}{L_{0}}\right]}^{-\frac{1}{2\alpha }}$$
where x is the drawn length. L(x) is the heating length of the taper zone in the optical fiber drawing process. α is the linear change factor, which is 0.1 in this paper. The parameters of the OM tapered transition area are shown in Figure 2. The initial cladding radius $r_{0}$ = 62.5 μm, and the initial core radius $r_{c0}$ = 5 μm. According to Equation (4), the initial core radius $r_{c0}$ will first approach zero before $r_{0}$ = $r_{w}$. From the end of the tapered transition area to the uniform waist area, the core has been integrated with the cladding, which is called ‘No core area’, denoted as $Z_{w1}$.

The relation between the length of the tapered transition area $l_{z}$ and the drawn length x can be expressed as [33]
(5)$$l_{z}\left(x\right)=\frac{1}{2}\left[x+L_{0}-L\left(x\right)\right]$$
where $L_{0}$ is the initial heating length, which is 3.8 mm wide of the heating unit. When the radius of the uniform waist area $r_{w}$ = 0.9 μm, the length $l_{w}$ = 10 mm. According to Equations (4) and (5), the overall length of the tapered transition area $l_{z1}$ = 22.3 mm, the length of inner core $l_{z2}$ = 19.4 mm, and the axial length of $Z_{w1}$ is $l_{z3}$ = 2.9 mm. According to the above calculation results, the tapered transition area has some similar structural parameters to the uniform waist area. Therefore, the influence of the tapered transition area should be considered in the process of analyzing the photoinduced thermal phase shift effect of OM.

According to the OM single mode transmission condition [32], it is considered that the pump light is no longer confined in the core when its radius $r_{c}$ in the tapered transition area decreases to about 0.35 μm. Before the core is reduced to this radius, the pump light at 980 nm is absorbed mainly in the core, and the temperature change caused by the heat generated in this part is denoted as $\Delta T_{1}$. The phase change is mainly caused by the change of temperature, so the phase change can be obtained by analyzing the heat distribution and diffusion. The approximate calculation expression of temperature rise in optical fiber is
(6)$$\Delta T=\frac{\eta \alpha lP}{C_{v}\rho \pi r^{2}}$$
where P is the intensity of the pumping light, l is the fiber length, and α is the loss coefficient of the pumping light in the fiber. The specific heat of SiO_2_ material is 741 J/(kg·°C), and the density ρ is 2320 kg/m^3^. The calculated area is equivalent to a cylinder of the same volume, and $\Delta T_{1}$ = 0.005 °C can be calculated from Equation (6). The area where the pump light begins to leak out of the core to the waist area is denoted as $Z_{w2}$. Taking the length of uniform waist area $l_{w}$ of OM as an example, the temperature rise in the $Z_{w2}$ area $\Delta T_{2}$ = 0.1 °C can be calculated by the equivalent calculation with the same method. According to the previous discussion, the uniform waist area of OM is the main heat producing area. According to the law of heat diffusion, the heat in OM is mainly diffused from the uniform waist area to the tapered transition area on both sides. Due to the symmetric structure of OM, half of the uniform waist area can be analyzed and recorded as ‘half waist area’. The calculation method is the same as above, and the temperature rise in this area is $\Delta T_{3}$ = 0.9 °C.

To calculate the axial heat diffusion, FEM is used for simulation. The initial ambient and cladding temperatures are set to zero. According to the above calculation results, the increased temperature ΔT is assigned to the corresponding region, and the above model is established in the FEM software. The established heat distribution model is shown in Figure 3. Considering only the solid heat transfer between optical fibers, the axial heat diffusion of OM is simulated.

According to the calculated temperature, the temperature of the ‘half waist area’ and ‘no core area’ of OM increases, which is the primary area with phase response occurring. To simplify the analysis, the two areas were combined and named ‘work area’. The temperature variation of other lumbar length regions can be calculated, and a similar simulation model can be established. At the same time (0.5 s), relevant simulation results such as the change of the average temperature of the ‘work area’ were obtained, as shown in Table 1.

According to the above calculation and simulation results, when the length of the waist region of OM increases, the temperature rise of OM caused by the photo-induced thermal effect will increase, resulting in a larger phase response amplitude of the OM phase modulator.

## 3. The Experimental Method and Results

The experimental setup is shown in Figure 4. The basic structure is a Michelson interferometer composed of a 2 × 2 coupler and two Faraday rotating mirrors (FRM). One arm of the interferometer is a ring-shaped piezoelectric ceramic (PZT) wrapped with an optical fiber, and the other arm is an OM. The signal light source of the system selects 1550 nm high-coherence light as the signal light of the interferometer, which is injected into the interferometer through the isolator. The 980 nm pump light was used as modulated light and heats the OM through a 980/1550 nm wavelength division multiplexer (WDM) to generate phase modulation. At the same time, the WDM can effectively prevent the reflected 980 nm light from entering the interferometer. The 980 nm pump laser is controlled by the corresponding driver, and the intensity modulation of the 980 nm pump light is achieved by applying different modulation signals on the driver. The interference signal output after phase carrier modulation can be expressed as [34]
(7)$$V=M+N\cos[C\cos\omega_{c}t+\phi_{s}\left(t\right)+\phi_{0}]$$
where C is the phase amplitude generated by the carrier modulation signal, $\omega_{c}$ is the angular frequency of the carrier modulation signal, M is the direct current term related to the light intensity, N is the coefficient related to the light intensity and the visibility of interference fringes, $\phi_{s}\left(t\right)$ is the additional modulation (sound pressure, vibration, or other phase modulation) response phase signal, and $\phi_{0}$ is random phase noise caused by the environment (usually a low frequency signal). The interference light is input into the computer through the photoelectric converter and the signal collector for demodulation, and the response phase signal $\phi_{s}\left(t\right)$ can be demodulated.

### 3.1. Phase Modulation Efficiency and Performance Optimization Test of OM Phase Modulator

The demodulation system is a PGC-Atan scheme based on external modulation. In this section, the PZT covered with fiber is the external modulator that generated the modulated carrier wave. The OM phase modulator is modulated by pump modulating light at 980 nm with different modulating frequencies and amplitudes. The performance was tested by demodulating the phase response signal. When a 6.25 kHz modulation signal is applied to the OM modulator, the demodulated modulation signal is shown in Figure 5.

The experimental results show that the OM phase modulator has a good phase modulation response. The phase modulation response signal is consistent with the frequency of 980 nm pump light modulation signal, which can achieve good phase modulation function. Additionally, the system can demodulate the corresponding phase modulation response signal. The improved flame scanning method was used to make OM with different structural parameters. The length of the uniform waist area was 10 mm, 12 mm, and 15 mm, and the diameter of the uniform waist area was 1.4 μm, 1.6 μm, and 1.8 μm, respectively. The nine samples of these structural parameters were studied as phase modulators.

To test the modulation bandwidth of the OM phase modulator, the modulated signal amplitude and bias voltage of the 980 nm pump light are kept. The average optical power of the modulated pump light is about 40 mW, and the modulation frequency (50 Hz~10 kHz) is changed. The frequency response characteristics of the OM phase modulator are tested and shown in Figure 6. With the increase of the frequency of the 980 nm pump light modulated signal, the amplitude of the phase modulation response of the OM phase modulator decreases, which is inversely proportional to the modulation frequency.

To further test the modulation performance of the OM phase modulator, the modulation response efficiency was also tested. The signal frequency of 980 nm pump light modulation is kept. Taking 1 kHz modulation frequency as an example, the amplitude of the modulation response of the OM phase modulator is measured when the modulation amplitude of 980 nm pump light is different. The test results are shown in Figure 7. The amplitude of modulation response of the OM phase modulator is proportional to the amplitude of 980 nm pump light (signal amplitude), and the linear response characteristic is good.

By comparing the result, it can be found that decreasing the uniform waist diameter and increasing the uniform waist length of OM within a reasonable range can make its phase response amplitude move up, which is consistent with the theoretical analysis.

The experimental results show that when the uniform waist area diameter of OM is 1.4 μm and the length is 15 mm, the phase response amplitude of the OM phase modulator can reach more than 1 rad within 10 kHz, which is enough to meet the requirements of sensing and other applications.

### 3.2. Test of Simulating Acoustic Signal Demodulation

In this section, the external modulator that generated the modulated carrier wave was changed. The OM phase modulator structural parameters are 1.4 μm diameter and 15 mm length of the uniform waist area, which is used as the phase carrier generation external modulator instead of the PZT covered with fiber. The PZT covered with fiber is used to simulate the underwater acoustic signal to be measured. In the experiment, the modulation frequency of OM modulator is 6.25 kHz. By adjusting the modulation amplitude of 980 nm pump modulated light, a stable and effective phase-carrier modulation signal can be generated.

The signal generator is used to apply the 1 kHz modulation signal on the PZT, and the phase response signal $\phi_{s}\left(t\right)$ demodulated by the system is shown in Figure 8a, and its spectrum diagram is shown in Figure 8b.

As shown in Figure 6, greater modulation depth can be obtained by reducing the modulation frequency. The modulation frequency is reduced to 2 kHz. Additionally, the true modulation depth is calculated from the interference signal to modify the PGC demodulation algorithm. Thus, the resonant peak is reduced. At the same time, the modulation frequency of PZT was adjusted to a lower frequency to test the low frequency demodulation capability of the system. Taking 170 Hz as an example, the demodulation results are shown in Figure 8c, d. There is a slight drift in the waveform of Figure 8c, which we think is due to ambient airflow. Above all, the waveform of the sinusoidal signal with the same frequency as the modulated signal can be demodulated, and the signal response is obvious on the spectrum diagram.

## 4. Discussion

In this paper, an all-optical phase modulator based on the photo-induced thermal phase shift effect is proposed. Starting with two important structural parameters of OM, a conclusion is proposed by combining theory with an experiment: increasing the length and decreasing the diameter of the OM uniform waist area can improve the phase response amplitude of OM modulator. However, this conclusion only applies to a reasonable range. When the uniform waist area diameter of OM is reduced to less than 1μm, the optical energy density in the uniform waist area decreases due to leakage, resulting in a decreased phase response. In a practical application, decreasing the waist diameter and increasing the waist length of OM will increase the difficulty of the production process and make OM fragile enough to easily break under external influence. Furthermore, increasing the length of the OM uniform waist area will lead to an increase in the tapered transition area length at the same time. Therefore, it’s limited to optimizing the OM modulator based on the structural parameters of OM. Thus, we will explore other methods to improve the performance of the OM modulator to achieve the optimum modulation depth for PGC demodulation. The modulation performance of the phase modulator can be improved by increasing the absorption rate of OM to the pump light, such as vanadium doping [35,36].

According to the demodulation result of simulating acoustic signal demodulation below 1 kHz, and comparing the demodulation spectrograms in Figure 8b,d, it can be seen that there are low-frequency noises. Although the phase noise caused by the light source internal modulation method is avoided fundamentally, the low frequency detection properties of the system will be affected by the external noise and the thermal noise generated by electronic equipment and the air flow. To improve the SNR by reducing the system noise and avoid environmental disturbances, further research on the packaging design and noise suppression of the modulator is needed.

## 5. Conclusions

In summary, a novel phase generated carrier device based on the photo-induced thermal phase shift effect is proposed in this paper. The influence of the structural parameters of OM on the effect was analyzed by combining theory with an experiment. Based on the research of this mechanism, the structural parameters of OM are optimized, and an optical microfiber all-optical phase modulator with larger modulation bandwidth and modulation depth is obtained. When the length of the OM uniform waist area is 15 mm and the diameter is 1.4 μm, the modulation amplitude of the phase modulator can reach 1 rad. It can meet the requirements of the PGC demodulation system. Carrying out related demodulation tests, the simulated underwater acoustic signal below 1 kHz can be demodulated successfully. Compared with other all-optical phase modulators, the optical microfiber all-optical phase modulator proposed by us can meet the requirements of the PGC demodulation system of fiber optic hydrophones.

## Figures and Tables

**Figure 1 nanomaterials-11-02215-f001:**
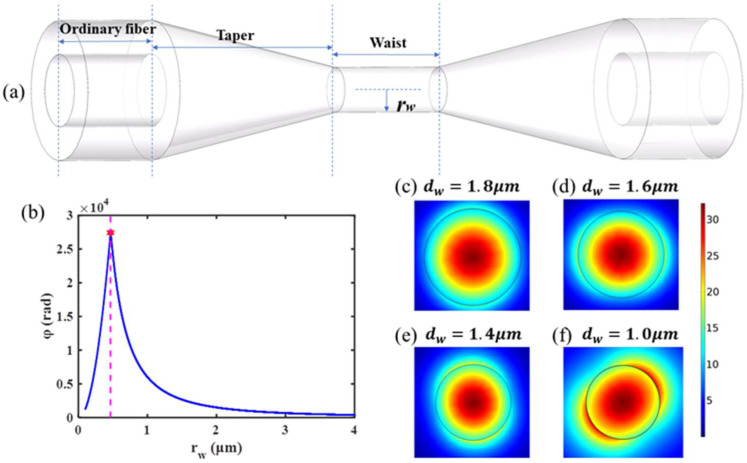
(**a**) The schematic diagram of the optic microfiber. (**b**) The relation between the phase response of OM induced by the photo-induced thermal phase shift effect and the radius $r_{w}$. (**c**–**f**) Energy distribution of the uniform waist area of OM with different diameters.

**Figure 2 nanomaterials-11-02215-f002:**
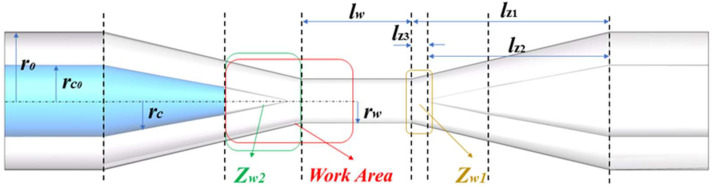
Annotation diagrams of OM-related parameters used in the calculations of this section.

**Figure 3 nanomaterials-11-02215-f003:**
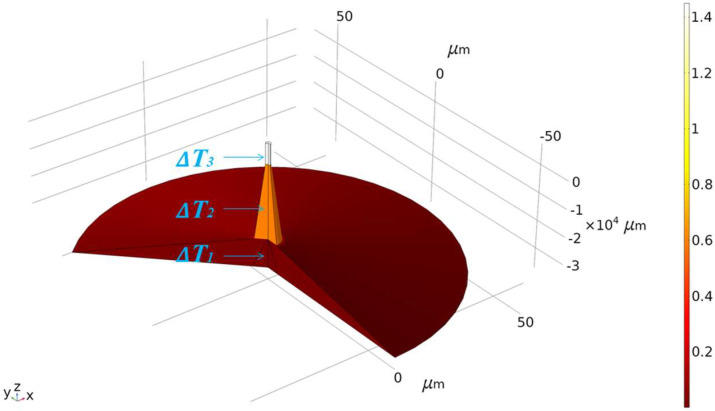
The heat distribution model diagram established by using FEM, and the calculated temperature change ΔT in the corresponding area was marked.

**Figure 4 nanomaterials-11-02215-f004:**
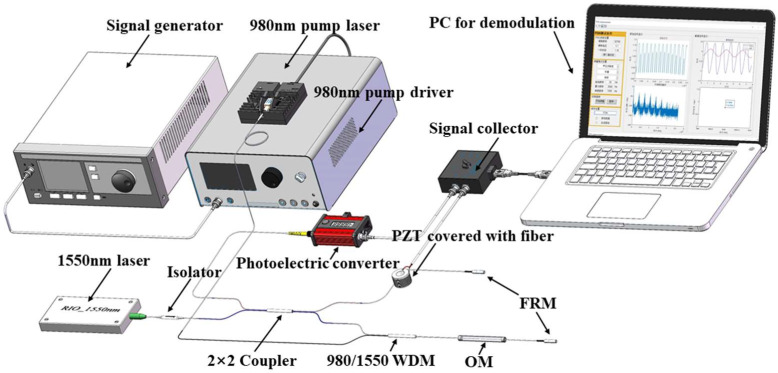
The setup of experiment.

**Figure 5 nanomaterials-11-02215-f005:**
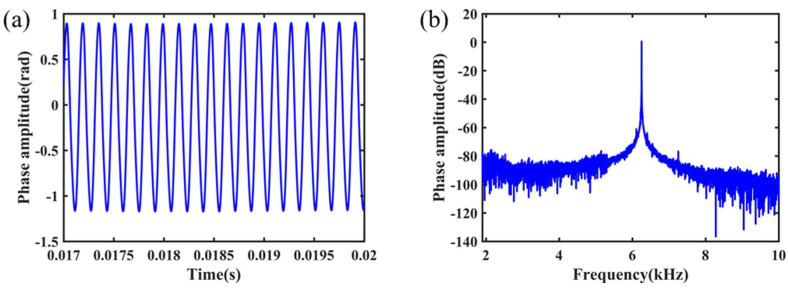
The modulation frequency of OM phase modulator is 6.25 kHz; the system demodulated the signal in (**a**) time domain and (**b**) frequency domain.

**Figure 6 nanomaterials-11-02215-f006:**
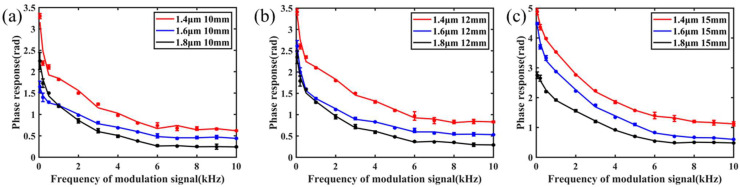
Frequency response of the OM phase modulator with different structural parameters; the lengths are the same as (**a**) 10 mm, (**b**) 12 mm, and (**c**) 15 mm.

**Figure 7 nanomaterials-11-02215-f007:**
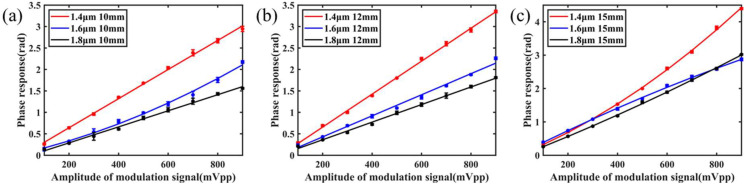
Amplitude response of the OM phase modulator with different structural parameters; the lengths are the same as (**a**) 10 mm, (**b**) 12 mm, and (**c**) 15 mm.

**Figure 8 nanomaterials-11-02215-f008:**
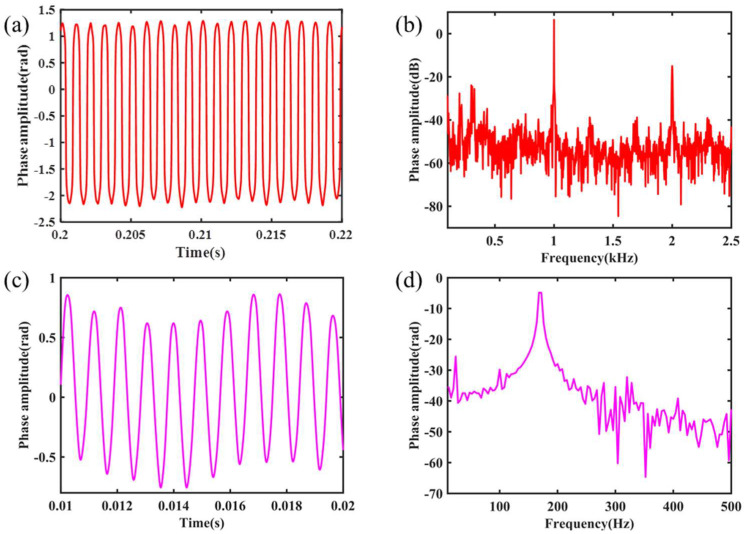
The demodulation signal of the 1 kHz simulating acoustic signal in (**a**) time domain and (**b**) frequency domain; the 170 Hz simulating acoustic signal in (**c**) time domain and (**d**) frequency domain.

**Table 1 nanomaterials-11-02215-t001:** The value of the relative change in ‘work area’ within 0.5 s.

Length of the Uniform Waist Area (mm)	Average Temperature Increase of the ‘Work Area’ (°C)	Average Temperature Decrease of the ‘Work Area’ (°C)	The Percentage of Decrease as Increase at ‘Work Area’
10	0.132	0.009	6.8%
15	0.602	0.022	3.62%
20	1.234	0.030	2.43%

## Data Availability

Not applicable.

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
