# Peer review of "Optical Microfiber All-Optical Phase Modulator for Fiber Optic Hydrophone"

_nanomaterials, 2021, doi:10.3390/nano11092215_

Round 1

Reviewer 1 Report

The manuscript entitled “Optical Microfiber all-optical phase modulator for fiber optic hydrophone” reported a novel phase generated carrier device based on the photo-induced thermal phase shift effect in an optical microfiber. The dependence of the device performance on the structural parameters of the optical microfiber was theoretically and experimentally investigated. The study presented herein is quite comprehensive and could be interested to readers in the field of acoustic sensing. I have a few comments for the authors to consider:

  • There are a few grammar mistakes in the abstract section, please correct them.
  • Figure 1(d) gives the energy distribution within optical microfibers with different diameters d However, dw was not defined in the context. Please be consistent with the parameter, either radius or diameter.
  • The frequency domain results shown in Figure 8 indicate that the SNR of the demodulated signal deteriorated in the low-frequency regime. Can the author provide a figure showing the SNR of the demodulated signal as a function of frequency?

Reviewer 2 Report

Manuscript No:  nanomaterials-1320854

Title:  Optical Microfiber all-optical phase modulator for fiber optic hydrophone

Authors:  Minwei Li 1,2,4, Yang Yu 2,4,*, Yang Lu 3, Xiaoyang Hu 3, Yaorong Wang 2,4, Shangpeng Qin 1,2,4, Junyang Lu 1,2,4, Junbo 4 Yang 2, and Zhenrong Zhang

  1. Overview
  2. In this manuscript the authors report on theoretical, simulation and experimental work on an optical microfiber all-optical phase modulator based on the photo-induced thermal phase shift effect to be used as a phase carrier generation component.
  3. The contents are expressed clearly; the manuscript is well organized and written in reasonable English.
  4. The authors have acknowledged recent related research.
  5. As long as my knowledge, the work presented is original and it is correct from a scientific point of view.

  1. Detailed analysis

Abstract: It is too long: It must be clear, objective and self-explanatory. State what have you done, how did you do it, the quantitative results you got and the novelty of your work. Please make it as synthetic as you can.

  1. Introduction: too lengthy - authors could be more direct.

  1. The effect of structural parameters of OM on photo-induced thermal phase shift

- Please find a shorter section title

- Section provides an interesting and detailed approach to the device operating principles and there are up to date references.

  1. The Experiment and results: In advise “3. Experimental Results”

- Section provides a systematic presentation of the simulation results.

- This section should be re-named:   “3. Simulation Results and Discussion”

- It seems  that figures 6 and 7 are not in template style.

- Error bars are missing in these figures  - experimental data

  1. Overall assessment

The work reported presents reasonable utility for supplementary studies and developments in the field.

In my opinion it can be published after corrections.

  1. Review Criteria
  2. Scope of Journal

Rating: Medium

  1. Novelty and Impact

Rating: Medium

  1. Technical Content

Rating: Medium

  1. Presentation Quality

Rating: Medium

Reviewer 3 Report

The manuscript reports on a novel fiber-optic phase modulator and its use in phase-generated carrier system. The working principle of the modulator is based on the heating of a tapered fiber section due to the absorption of an additional pump signal. By controlling the intensity of the pump signal, it is possible to modulate the phase shift of the target optical signal. The work is organized well, contains a rich literature review and is put into a proper context. The reported results might be applied in sensing systems designed to operate in harsh environments, when it is problematic to use conventional electro-optical modulators. I have, however, several comments, that, I hope, the authors will find useful.

  1. It is not straightforward to see how figure 1 b follows from eq. (2). I can assume that A_eff(r_w) dependency causes this non-monotonous behavior. Please explain this point more clearly, as it is one of the crucial aspects of the whole work.
  2. In experimental section, since two modulators are used, it is advisable to clearly indicate the functions each of them performs. In section 3.1 PZT-based modulator is used as an additional (auxiliary) modulator, while the OM modulator acts as a target modulator (the phase shifts that it induces are to be measured). In section 3.2 the situation is reversed - after the properties of OM modulator are studied, an experiment demonstrating feasibility of OM modulator use as an additional modulator is also carried out. Direct statement of each modulator role in each experiment will help reader less familiar with PGC technique better understand the work.
  3. I would also advise to indicate which PGC demodulation algorithm was used to obtain the phase signals, since doing so will help the reader better understand the work and the potential sources of its limitations (see points 4 and 5 below).
  4. Regarding the results in section 3.2, I would say that their presentation is a bit misleading, since only the main harmonic is shown in figure 8 b and 8 d. However, by looking at signals in figure 8 a and 8 c, it can be seen that the demodulated signals are subject to nonlinear distortions, which would be seen more clearly if horizontal scales in figures 8 b and 8 d were expanded to show at least 5-10 harmonics.
  5. Another question related to the final results is the following - what are the possible sources of such nonlinearity. Is it performance of OM modulator? Typically, such nonlinearity is due to drift of the modulator efficiency (when practical modulation depth is different from the one assumed in the demodulation algorithm). If so, what can be done to stabilize OM modulator characteristics? Or, maybe using another demodulation algorithm (there actually is a vast number of PGC algorithms, some of which are able to estimate the modulation depth directly from the interference signal) can lead to better results.
  6. Finally, the English usage must be revised.

Round 2

Reviewer 3 Report

The authors have made necessary corrections to the manuscript. The only remaining point is related to equation numbering - there are two eqs. with number (2) and the following numbers must be corrected.
